# A self-supervised learning approach for high throughput and high content cell segmentation

Van K. Lam, Jeff M. Byers, Michael C. Robitaille, Logan Kaler ⓘ, Joseph A. Christodoulides & Marc P. Raphael ⓘ ✉

In principle, ML/AI-based algorithms should enable rapid and accurate cell segmentation in high-throughput settings. However, reliance on large training datasets, human input, computational expertise, and limited generalizability has prevented this goal of completely automated, high-throughput segmentation from being achieved. To overcome these roadblocks, we introduce an innovative self-supervised learning method (SSL) for pixel classification that does not require parameter tuning or curated data sets, and instead trains itself on the end-users' own data in a completely automated fashion, thus providing a more efficient cell segmentation approach for high-throughput, high-content image analysis. We demonstrate that our algorithm meets the criteria of being fully automated with versatility across various magnifications, optical modalities, and cell types. Moreover, our SSL algorithm is capable of identifying complex cellular structures and organelles, which are otherwise easily missed, thereby broadening the machine learning applications to high-content imaging. Our SSL technique displayed consistently high F1 scores across segmented cell images, with scores ranging from 0.771 to 0.888, matching or outperforming the popular Cellpose algorithm, which showed a greater F1 variance of 0.454 to 0.882, primarily due to more false negatives.

Today, 384 and 1536 well plate layouts are quickly replacing the past norm of 96 well plates and becoming commonplace in high-throughput assay applications, such as drug impact and toxicity assessment, expanding the range of cell lines, treatments, phenotypes, and surfaces for a given experimental dataset[1–3]. To keep pace, the application of machine vision for cell segmentation has grown remarkably to avoid time-consuming and intensive manual equivalents. Automated cell segmentation is seen as a cornerstone for high-throughput studies, highlighting the need for software that can facilitate the accurate assessment of cellular morphology. The applications are broad for automated cell segmentation due to the need for swift and precise analysis in high-throughput contexts for drug development, real-time gene-expression tracking, and cell therapy[4–7] to name a few. Despite numerous reports of success with current automated segmentation tools, the application of deep learning in cell segmentation still encounters challenges, including a significant demand for data, the need for human intervention and biases within, limited generalizability, and specialized training.

Deep convolutional neural networks (CNNs) have gained prominence in deep learning-based cell segmentation, showcasing the efficiency and accuracy of UNET[8–12] and Mask R-CNN[13] in cell and nucleus segmentation. This technique interprets segmentation as a task of identifying cell boundary pixels, compelling the model to produce a "distance map" for each pixel to estimate its likelihood of being at a cell border. In high-throughput assays, a diverse dataset capturing a broad spectrum of cell morphologies is crucial for CNNs to accurately segment individual cells. However, the extensive requirement for data training and manual annotation restricts the application of CNNs in high-throughput studies with limited replication and presents a significant hurdle for researchers who are unable to expand their studies without additional training data. A recent study highlights that effective CNN model training requires a substantial, high-quality pre-training dataset consisting of 1.6 million cells, presenting the challenge of applying CNN methods in segmenting specialized high-throughput studies due to the sheer scale of data needed[14]. Other techniques like pixel classification are preferred for images with low cell density when it involves labeling each pixel in an image as a distinct class (e.g., cell versus background)[15–18]. High fidelity segmentation is essential to further downstream tasks, such as declumping and organelle identification using post-processing algorithms like watershed[19], active contour[20], or threshold-based segmentation[21].

US Naval Research Laboratory, Washington, DC, USA. ✉e-mail: marc.p.raphael.civ@us.navy.mil

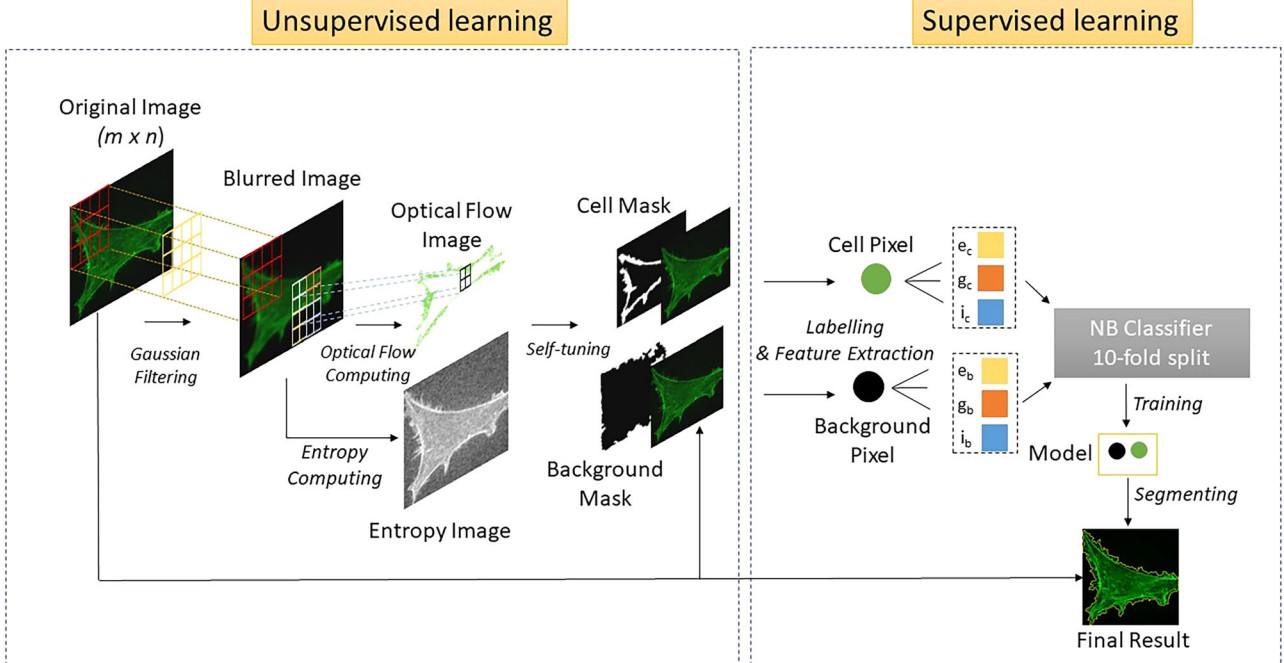

**Fig. 1 | Flow chart of self-supervised learning segmentation.** The process starts with generating a blurred (m × n) image, then applying unsupervised learning via optical flow (OF) to label cell and background (bg) pixels. Supervised learning is performed by extracting labeled (cell or background) pixel features from the image, including entropy (e), gradient (g), and intensity (i). A Naïve Bayes classifier with the ten-fold split is used for training. The classification model is exported and applied to the original image. This method is repeated with each image, ensuring an adaptive process in which every image has a uniquely associated classification model.

According to the 2018 Data Science Bowl, the top three performing CNN models, namely U-Nets, fully connected feature pyramid networks, and Mask R-CNN demand high computational expertise for segmentation tasks, resulting in a significant obstacle to biologists who are the primary end-users of these models[9]. Furthermore, it is of paramount importance to expand the training datasets and enhance their ability to generalize across different studies that go beyond competition scenarios. Given that scientists frequently resist expanding extensive imagery datasets for additional training, an innovative strategy in bio-imaging platforms adopted by CNNs, such as Cellpose (CP) 2.0[22], involves incorporating a "human-in-the-loop" feature to aid in segmentation. This feature enables users to adjust crucial cell parameters manually or undertake the segmentation of specific cells of interest for a more targeted training set. Nevertheless, this approach introduces additional labor and time, diverging from the goal of completely automated high-throughput imaging. Furthermore, users may have to offer numerous manual segmentation references for each condition in the high-throughput assay, potentially increasing bias.

Many high-throughput imagers are now equipped with high magnification, immersion objectives for high-resolution imaging (e.g. objectives with 50X–60X magnification and high numerical apertures), giving users far more intracellular information than the more traditionally utilized 10X–20X magnifications can provide. This synergy of high-throughput and high-resolution imaging can significantly enhance the comprehensive understanding of cell phenotype, covering aspects, such as sub-cellular structure[23,24], molecular expression[25], and organelle distribution[26]. For instance, high-resolution imaging's potential enables scientists to quantify cell membrane convexity and concavity, offering insights into the interactions between cytoskeleton and substrate environment[27]. Despite these advances, current studies often overlook the meticulous analysis of fine cell structure due to the limitations of study size and a lack of annotated training data required for high-content automated segmentation. Consequently, biomedical researchers are frequently compelled to choose between image resolution and sample size. The future of cell imaging must bridge the gap between high-resolution and high-throughput methodologies by advancing toward their integration, making fully automated segmentation an essential analytical tool. With this capability, researchers can unlock a far more comprehensive understanding of cellular behavior captured from detailed structural nuances of vast datasets, generated from imaging thousands to millions of cells across various conditions, materials, and time points.

To address these challenges, we present a self-supervised learning (SSL) methodology for pixel classification tasks, aimed at automating cell segmentation for both high-content and high-throughput data studies. This user-friendly and robust approach is completely automated, eliminating the need for dataset-specific adjustments and data annotation in the training process. Our previous segmenting method utilized optical flow (OF) vector fields generated between consecutive images from live cell time series data sets as an inherent data structure to self-label training data, thereby creating a robust and truly automated cell segmentation technique that performed well against contemporary methods[18]. Building on this, we demonstrate a method *applicable to both live or fixed cell imagery*, achieved by blurring a single image and then employing SSL techniques to both the original and blurred images (Fig. 1). Specifically, we employ a Gaussian filter from an original input image, then calculate OF between the original and blurred image. OF vectors are used as a basis for self-labeling pixel classes ("cell" vs "background") to train an image-specific classifier. We demonstrate that this SSL approach achieves complete automation while also capturing high content information with a robustness that extends across a range of magnifications and optical modalities. As a result, we present this algorithm as an enabling technology that applies across disciplines, from exploratory cell biology to targeted biomedical research studies to biomanufacturing quality control. This work focuses on pixel-based segmentation for cell confluency calculations and intracellular phenotyping investigations. As an additional feature, we demonstrate that our segmentation results can be utilized as inputs into cell declumping algorithms with high-fidelity results for non-adherent cells. Declumping of adherent cells will be the focus of future work.

**Fig. 2 | Self-supervised learning segmentation approach for a range of cell types, optical modalities, and magnifications.** Phase contrast image of **a** MDA-MB-231 (10X objective), bright-field image of **b** MDA-MB-231 (40X objective), (**c**) DIC image of MDA-MB-231 (i) (20X objective) and *S. cerevisiae* (ii) (63X objective), (**d**) IRM image of Hs27 (40X objective), (**e**) epifluorescence images of (i), (ii) A549 cells with lifeAct (GFP-actin conjugate, 100X objective) and (iii, iv) MDA-MB-231 (*F*-actin (red) and vinculin (green) expression, 63X objective). Yellow borders show SSL segmentation results. Scale bar **a**: 50 μm, **b**: 10 μm, **c.i**: 25 μm, **c.ii**: 5 μm, **d**: 20 μm and **e**: 20 μm.

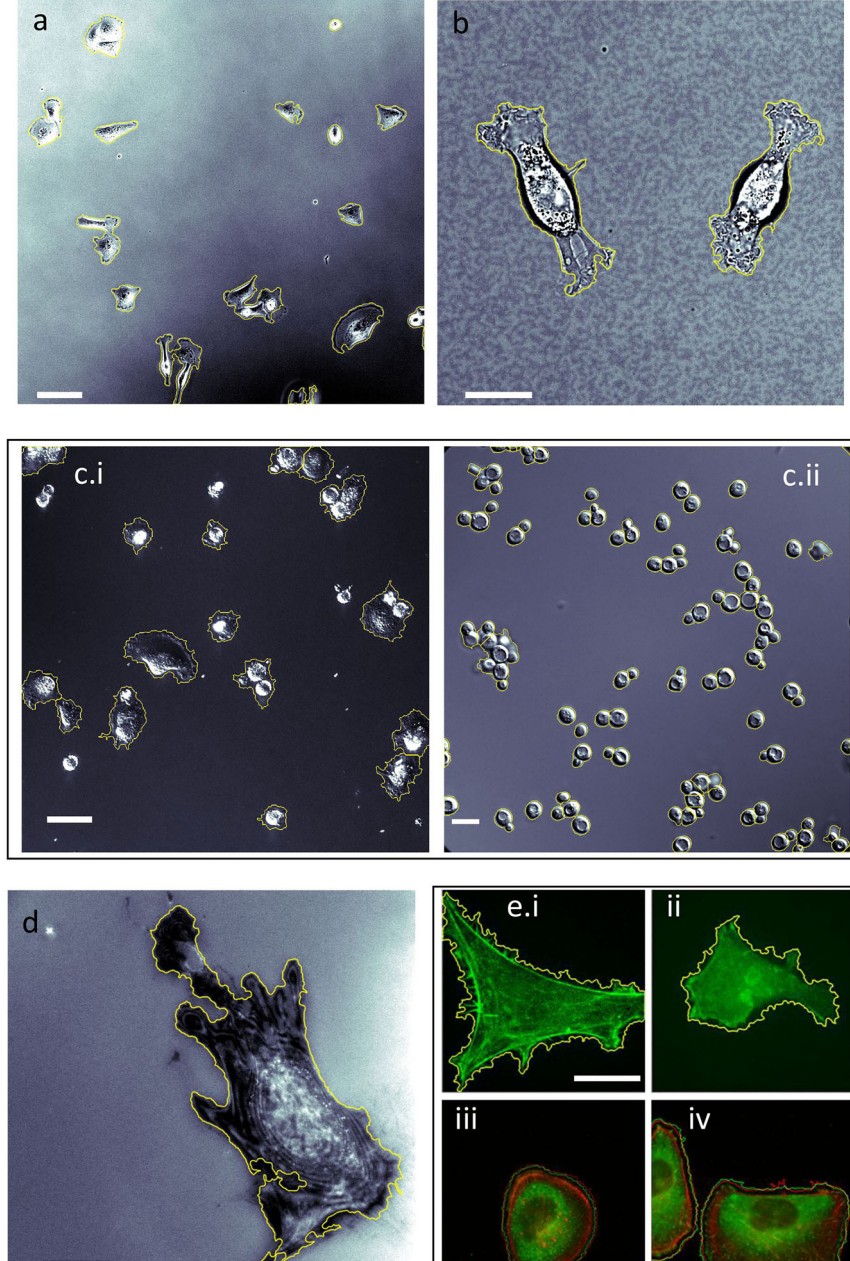

## Results

### SSL approach for segmenting diverse microscopy modes across multiple cell types

We first aim to highlight the compatibility of our SSL code with high-throughput imaging in that robust cell segmentation takes place from single images in the absence of both curated training libraries and manual parameter settings. Our SSL method demonstrated versatility by working at different resolutions and with different microscopy modalities spanning from label free to fluorescent imagery (Fig. 2a–e). Representative images segmented by SSL varied from mammalian cell types to fungi, were processed in a completely automated fashion. Label-free examples includes breast cancer cells MDA-MB-231 imaged by phase contrast at 10X (Fig. 2a) and bright-field at 40X (Fig. 2b); differential interference contrast (DIC) images of MDA-MB-231 at 63X (Fig. 2c.i) and *S. cerevisiae* (Fig. 2c.ii); human fibroblast Hs27 imaged by interference reflection microscopy at 40X (Fig. 2d); epifluorescence images of GFP-labeled A549 cells (Fig. 2e.i and ii) at 40X (previously analyzed as live cell imagery in Robitaille et al. and

Christodoulides et al.[17,28,29]), F-actin and vinculin-labeled MDA-MB-231 cells (Fig. 2e.iii and iv) at 63X. The combined segmentation of both structures—*F*-actin (in red) and vinculin (in green)—appears yellow, allowing for further co-localization analysis of these structures. An additional declumping step was employed in Fig. 2c.ii after segmentation as described in more detail below.

### SSL for segmenting diverse fluorescent channels across multiple cell treatments

Fluorescent Hs27 cell segmentation was performed on epi-fluorescent and confocal imagery with the cells cultured on different surfaces and stained with DAPI, phalloidin, and anti-Vinculin antibodies (Fig. 3). In a completely automate fashion, SSL achieved robust cellular segmentations across different stains: DAPI (Fig. 3a.i,iv), phalloidin (Fig. 3a.ii, b.i), and vinculin-antibody (Fig. 3a.iii, b.ii)); different modalities (epifluorescence vs confocal microscopy); and different resolutions (20X vs 63X). The nuclei were segmented with robust, clear borders, though in some instances, double nuclei were

**Fig. 3 | Self-supervised learning segmentation across different cell culture methods, resolutions, and cellular structures.** Fluorescent images of Hs27 stained for nucleus (DAPI), cytoskeleton (*F*-actin), and focal adhesions (vinculin) cultured on **a** glass-bottomed petri dishes using a 20X objective (epi-fluorescence) and **b** 63X confocal images of Hs27 cultured on a glass-bottomed petri dish stained for *F*-actin and vinculin. Scale bar **a**: 50 μm, **b**: 10 μm.

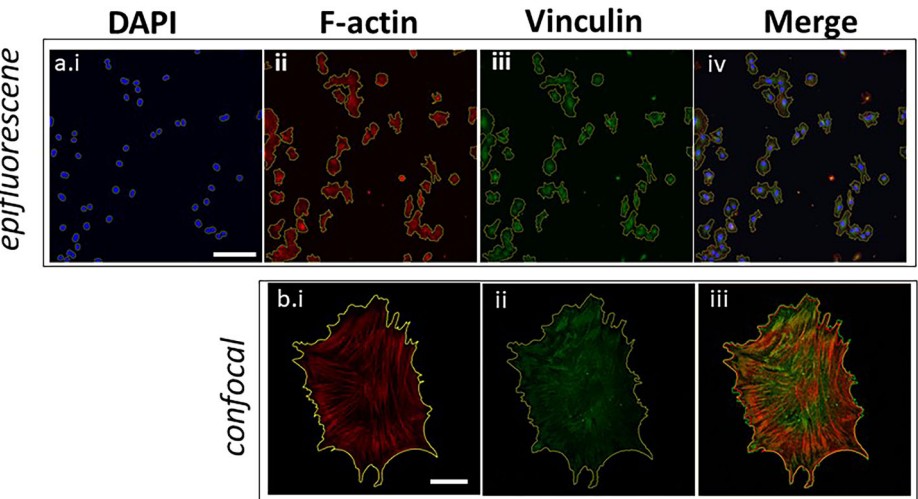

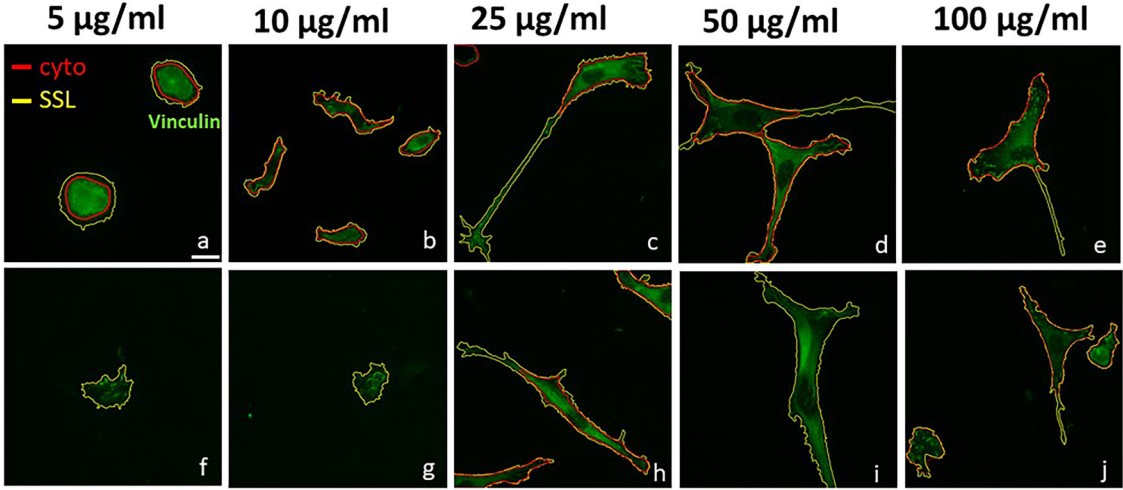

**Fig. 4 | Self-supervised learning segmentation approach in studying the effects of fibronectin concentration on Hs27.** Fluorescent images of Hs27 stained for focal adhesion by anti-vinculin antibodies (green) cultured on glass-bottomed petri dishes of which surface was coated with different fibronectin concentration using 40X objective. SSL results are outlined in yellow and Cellpose in red. Scale bar **a–j**: 20 μm.

observed which could be explained by cell division (Fig. 3a.i). Most individual cells were successfully segmented by SSL via *F*-actin and Vinculin structures across different imaging setups, demonstrating the segmentation's adaptability and precision at various resolutions without the need for curated pre-training data sets. In addition, membrane morphologies unique to a given stain could be distinguished upon merging these segmentations (Fig. 3b.iii).

## SSL segmentation approach for quantifying phenotypic stimuli

Given that most high-throughput experiments are performed on fluorescent data across a variety of treatment conditions, we conducted an analogous experiment by compiling images of fibroblast Hs27 plated atop surfaces treated with different fibronectin concentrations in a single executable run. These surfaces were drop coated with fibronectin of 5, 10, 25, 50, 100 μg/ml, fixed, and stained with anti-vinculin antibodies, then imaged under fluorescence microscopy using a 40X objective. Images of Hs27 cells cultured on surfaces treated at different fibronectin concentrations were loaded and processed with SSL in one shot, ensuring the initial self-tuning values for background and cell pixel labeling during unsupervised learning remained consistent.

As in previous examples, the need for a training dataset or tuning parameters was eliminated, enabling wide-ranging phenotype segmentation in an automated fashion. The entire segmentation process for this particular experiment was completed in 165 s on the laptop computer described above, efficiently processing 15 images throughout five conditions. In parallel, the same images were analyzed using the CP cyto2 model, with manual input for the object diameter set to 150 pixels, which was effective in most cases. Figure 4 presents a direct comparison between the segmentation results of CP (in red) and SSL (in yellow). Our SSL approach consistently delivered robust segmentation outcomes for Hs27 cells, requiring no manual intervention, even with the variations in cell shapes induced by fibronectin treatments. Conversely, the CP model sometimes failed to identify the elongated trailing edges of fibroblasts (Fig. 4c, e, h, j) or missed entire cells (Fig. 4i). Hs27 morphology varies dramatically between low and high fibronectin concentration coated on the cultured surface. Therefore, an adaptable segmentation approach like SSL can ensure that every image has a uniquely associated classification model. Indeed, SSL had no difficulty segmenting the completely unique phenotypes associated with low fibronectin treatments of 5 or 10 μg/ml (Fig. 4a, b, f, g) alongside higher concentrations without the need for a pretrained dataset.

## SSL for segmenting fine-detail cellular structure

The robustness of SSL in segmenting fine-detail cellular structures represents a significant advancement in high-resolution fluorescent imaging (Fig. 5). Our SSL methodologies exhibit a remarkable capacity

for accurately delineating intricate cellular components, a task that is essential for detailed biological analysis and interpretation. In fact, these structures are not easily seen by eye without contrast enhancements (Fig. 5 a.iii, b.iii and c.iii) and were often missed by CP. This precision in segmentation is particularly vital when dealing with complex cellular

structures like filopodia (Fig. 5a.iii and c.iii) and lamellipodia (Fig. 5b.iii), where the distinction between different cellular components can be subtle yet critical. To ensure the Fig. 5 results were not due to a lack of CP training data, we further compared results after implementing the CP human-in-loop feature (CP-HiL), which enables user-curated training sets to be incorporated into the model. A fluorescent dataset of Hs27 cells stained for F-actin, imaged by 10X epi-fluorescence and 63X confocal microscopy, was segmented using SSL, CP cyto (cyto2), and CP-HiL. For CP-HiL, seven 10X epi-fluorescence images and three 63X confocal images were used for the additional training.

The initial cyto2 segmentation failed to segment a number of cells on the 10X images (Fig. 6 a.i-iii), highlighted by white dotted boxes, prompting us to integrate the CP-HiL approach for manual corrections of misclassified segments. Though Fig. 6b.i shows that CP-HiL successfully identified cells missing from the CP method shown in Fig. 6a.i, CP-HiL failed to identify other cells or misinterpreted a single cell as two separate entities (Fig. 6b.ii,iii). Segmentation with SSL did not overlook any cells or falsely split single cells, though additional declumping algorithms for adherent cells will need to be added as a separate step in future work (Fig. 6c.iii).

As shown in Fig. 6, CP cyto2 and CP-HiL demonstrated higher fidelity in segmenting Hs27 images taken by confocal microscopy at 63X, but the two methods required manual tuning of the object diameter specific to each image. For instance, a user input diameter of 150 pixels was determined as optimal for CP methods to segment the Hs27 cells shown in Fig. 6a.iv and 6b.iv, but required diameters of 200 and 330 pixels for similar cell shapes shown in Fig. 6a.v and b.v, respectively. User adjustment of the CP diameter setting between 200 and 330 was also necessary for high fidelity segmenting of epithelial-like cells in Fig. 6a.vi. and b.vi. In contrast, our SSL approach maintained constant setting inputs across all images, resolutions, and cell shapes (Fig. 6c.i-vi), simplifying the segmentation process and eliminating the need for manual adjustments. In addition, and as detailed in Fig. 5, SSL consistently captured the fine structures in membrane extensions.

Given that the CP method provided similar results and required fewer human inputs comparing to CP-HiL, a quantitative comparison between CP and SSL was made. For this analysis, ten 10X fluorescent images were selected, and 80 cells within these images were randomly chosen for manual segmentation to create a "manual mask." The same cells were then

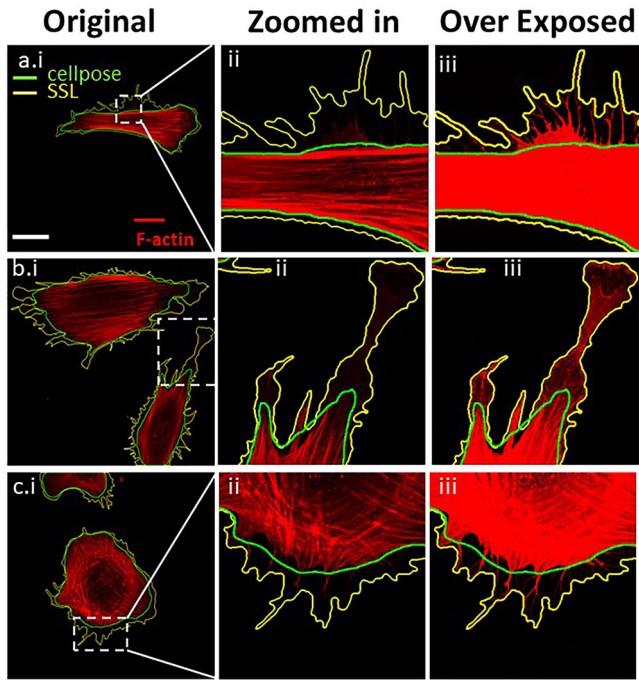

**Fig. 5 | Self-supervised segmentation approach on high-resolution images.** Confocal images of Hs27 stained with *F*-actin (red) for cytoskeleton at 63X objective (**a.i, b.i, c.i**). Images were zoomed in (**a.ii, b.ii, c.ii**) and contrast enhanced to depict structures of filopodia (**a.iii, c.iii**) and lamellipodia (**b.iii**). SSL results are outlined in yellow and Cellpose in green. Scale bar (white) **a.i**: 20 μm.

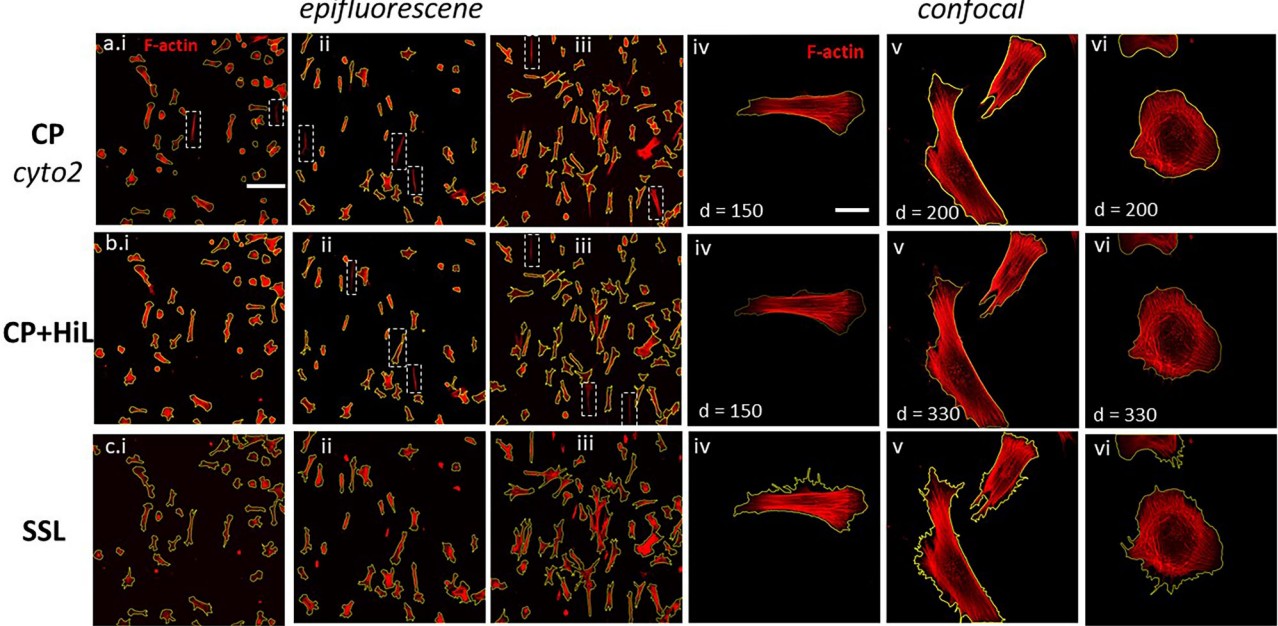

**Fig. 6 | Representative segmentation results on images of Hs27 stained with F-actin by (a) Cellpose (CP), (b) Cellpose with human-in-loop feature (CP-HiL), and (c) self-supervised learning (SSL).** Dotted-line boxes indicate cells or regions of interest that were missed during segmentation. Epi-fluorescence images were taken with a 10X objective and confocal images were taken at 63X objective. Scale bar **a, b, c** (**i-iii**): 50 μm; **a, b, c** (**iv-vi**): 10 μm.

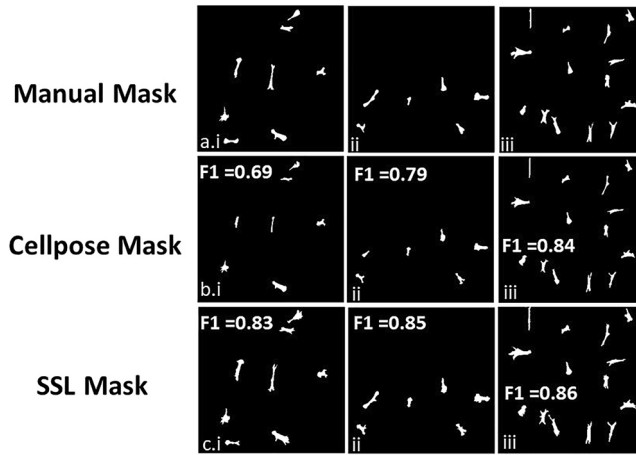

**Fig. 7 | Segmentation mask comparison of representative images.** Segmentation masks provided by (**a**) manual, (**b**) Cellpose, and (**c**) SSL segmentation and associated F1 scores.

segmented using both CP and SSL methods, generating "automated masks" for comparison. The F1 score was calculated by comparing the "manual mask" against the "automated masks" from each method. The average F1 score for SSL was 0.852 (S.D. = ± 0.017), surpassing the CP method's average F1 score of 0.804 (S.D. = ± 0.08). Representative masks from manual, CP, and SSL segmentation, along with their respective F1 scores, are illustrated in Fig. 6. The SSL method demonstrated stable F1 scores across all segmented images, ranging from 0.831 to 0.876. In contrast, the CP method's F1 scores varied more significantly, from 0.645 to 0.8815, primarily due to false negative instances (Fig. 7c).

## Quantitative analysis of SSL and CP performance on larger datasets

To broaden these results, a comparison between SSL and CP models was performed to evaluate segmentation performance across various biological objects, microscope modalities, image sizes, and fluorescent labels using hundreds of in-house and public data set images. Table 1 details the resulting F1 scores, processing times, and—because CP sometimes failed to segment entire images—the percentage of images successfully segmented. For the 10X epifluorescence and phase contrast Hs27 in-house datasets, the SSL F1 scores were comparable to CP cyto2, but with significantly faster processing times, saving 25 min across over 500 images. For the 40X Hs27 images, SSL achieved a higher F1 score of 0.831 and a faster processing time compared to CP cyto2, which also failed to segment approximately 12% of the images. A similar trend was observed for the 60X Hs27 images stained for *F*-actin and Vinculin in which the cyto2 and cyto3 F1 scores were 0.608 and 0.454, respectively, in contrast to a significantly higher F1 score of 0.758 for SSL. In addition, the CP models required more processing time while successfully segmenting only about half of those images.

When evaluating a Cell Paint dataset of Hoechst-stained nuclei that was also used to train the CP "nuclei" model29, CP outperformed SSL, achieving a F1 score of 0.953 with 100% segmentation in 58 min, versus a 0.873 SSL F1 score in 23.5 min. In general, the table shows that SSL demonstrates faster processing times and higher fidelity segmentation, particularly for the in-house Hs27 10, 40, and 60X datasets, compared to CP. Furthermore, as described above, SSL does not require that manual tuning of a diameter parameter for optimization.

Figure 8 presents representative image segmentations from Table 1 by SSL (Fig. 8a, c, e) and CP (Fig. 8b, d, f). In most scenarios, it is clear by eye that CP has a tendency toward false negatives, which lower its respective F1 scores in Table 1. The exception is with the nuclei segmentation, where SSL tended toward over-segmentation (Fig. 8e) compared to the "nuclei" model of CP (Fig. 8f). Additionally, SSL counted surrounding flow signals near neighboring nuclei, reducing segmentation precision relative to CP.

**Table 1 | Segmentation performance by SSL and an optimized Cellpose model for a range of imaging datasets**

| | Epifluorescence Hs2710X | | Phase contrast Hs2710X | | Epifluorescence Hs2740X | | Epifluorescence Hs2760X | | | Epifluorescence Nucleus | |
|---|---|---|---|---|---|---|---|---|---|---|---|
| Data source | in-house | | in-house | | in-house | | in-house | | | Cell Painting | |
| Stains | *F*-actin, Vinculin | | N/A | | Vinculin | | *F*-actin, Vinculin | | | Hoechst 33342 | |
| No. of images | 20 | | 565 | | 17 | | 67 | | | 189 | |
| Image Size | 2304 × 2304 | | 1344 × 1024 | | 1324 × 1024 | | 2262 × 2262 | | | 696 × 520 | |
| Segmentation technique | SSL | CP | SSL | CP | SSL | CP | SSL | CP | | SSL | *Cellpose* |
| Model* | - | cyto2 | - | cyto2 | - | cyto2 | - | cyto2 | cyto3 | - | nuclei |
| Diameter* | - | 100 | - | 100 | - | 150 | - | 400 | 400 | - | 30 |
| F1 | 0.888 | 0.882 | 0.771 | 0.777 | 0.831 | 0.818 | 0.758 | 0.608 | 0.454 | 0.873 | 0.953 |
| Times (mins) | 10.48 | 16.67 | 123 | 148.05 | 3.02 | 5.15 | 19.04 | 33.78 | 40.53 | 23.45 | 58.00 |
| % of exported segmentation images | 100% | 100% | 100% | 100% | 100% | 88.23% | 100% | 62.69% | 53.73% | 100% | 100% |

The table details information on data source, optical modalities, image size and required Cellpose inputs. Output metrics include F1 scores, processing times, and percentage of images segmented from the data set. Diameter (in pixels), *Only required by Cellpose.

**Fig. 8 | Representative image segmentation results from Table 1 by SSL and an optimized Cellpose-model. a** and **b** 10X Hs27 stained with Vinculin, **c** and **d** 60X Hs27 stained with *F*-actin and Vinculin, **e** and **f** nucleus stained with Hoechst. Scale bar: **a–b**: 50 µm, **c–d**: 25 µm, and **e–f**: 20 µm.

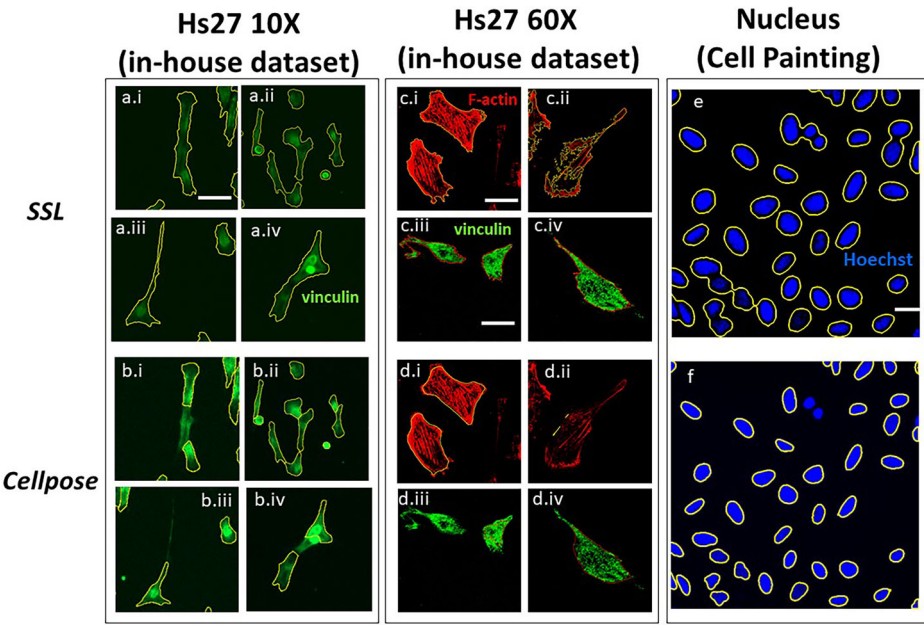

However, as noted above, CP was specifically trained on these images while still requiring the manual tuning of a diameter ($d = 30$) whereas SSL is self-trained and completely automated. Details of SSL and additional CP segmenting configurations, including models, sample size, percentage of successful segmentation, F1 score and time are given in Supplemental Note 3.

### Additional SSL declumping feature for non-adherent cells and nuclei

While this work is focused on automated segmentation for quantifying cell confluency and detailed intracellular features, we recognize that identifying individual cells within clumps is a naturally desirable extension of the current algorithm. Here, we demonstrate that the application of the watershed algorithm to SSL-segmented clumps enables the declumping of both *S. cerevisiae* and nuclei for applications from fundamental research to biomanufacturing quality control. We view this as proof of principle that SSL segmentation results can be utilized as downstream inputs for additional processing applications such as declumping. Figure 9 shows representative imagery and declumping results for both SSL and CP (cp_yeastPC_cp3 model) using our in-house 63X DIC images of *S. cerevisiae* ($n = 60$); and for the nuclei images of Table 1. SSL with watershed declumping preformed especially well on the yeast imagery compared to CP, which often failed to segment cells in the clump. We will continue to develop this feature for both non-adherent and adherent cell types in future work.

### Discussion and conclusion

High throughput in vitro imaging is now producing far more imagery—both in terms of number of images and diversity of imaging modalities—than even the most advanced deep learning algorithms can reliably segment. In addition, the increasing requirement of extracting fine cellular details from this imagery means that curated training data sets will need to grow exponentially at a rate that, in our view, is simply unsustainable. Machine learning and deep learning pretrained models, when trained on specific cell types or imaging techniques, often do not generalize effectively to other cell types or imaging modalities[22]. Challenges persist due to the inherent variability in cell phenotypes across different tissues and organisms that hampers the creation of universally applicable datasets. Transfer learning can offer a solution by conducting a pre-training model on a similar available dataset and applying it to an unseen target set. Nevertheless, limitations stemming from the pre-training dataset could lead to overfitting, where the model excels with familiar data but underperforms with

unexpected phenotypes in the target datasets[30]. This point becomes increasingly more relevant as the popular use of 384 and 1536 well plates in drug discovery applications expands the range of phenotypes. Such wide-ranging imagery datasets can require repeated deep-learning model retraining, making the training procedure increasingly labor and computationally intensive. The innovation of SSL streamlines this process, as each input image generates a specific model for its segmentation purpose, circumventing the pitfalls of over-generalization that often result in reduced accuracy.

The key enhancement to our previous SSL algorithms[17,18] is the applicability to single-frame data, eliminating the need for live-cell imagery and thus making the code more broadly applicable to include fixed cell imagery. The process of SSL begins with an unsupervised labeling of cell and background pixels by employing OF between the original image and a blurred version, thus eliminating the need for curated training data sets and ensuring data relevancy. This method proved effective in identifying intricate organelles and cellular structures regardless of magnification level or the use of external dyes (Figs. 2 and 3) as well as detecting structures that might be overlooked by the human eye (Figs. 4–6), thereby enhancing the generalizability of machine learning outcomes to high content imaging. The label library generated from the unsupervised step is employed in supervised learning training to classify pixels missed by the OF process. The supervised classification in this work leverages three features—entropy, gradient, and intensity—as they provide rich information at the cellular level but is readily expandable. Entropy, a measure of localized texture, is well-known for object edge detection[31], and recently applied in a wide range of disciplines like ecology and landscapes[32], evolutionary biology[33], and cancer research[34–36]. For in vitro microscopy, the inherent intracellular variability in organelle and cytoskeletal arrangements makes for enhanced texturing versus the background.

Another advantage of this SSL approach is traceability, which we define as ability to trace backward from the segmentation results to the image feature vectors that proved most influential. The extensive range of data features in deep learning - often requiring the tuning of millions of hyper-parameters—contributes to its "black box" nature such that scientists can be at a loss to understand the conclusions. This lack of transparency is particularly alarming in biomedical studies[37], where understanding the rationale behind a model's decisions is crucial for advancing discoveries in cellular activity, signaling pathways, or diagnostic procedures. Other pixel classification methods offer many more image features transformed by principal component analysis but their superposition can hamper insights[38,39]. In

**Fig. 9 | Representative segmentation results by SSL and the optimized Cellpose model on (a) 63X *S. cerevisiae*, and (b) nuclei stained with Hoechst.** From left to right, (i) original image to-be-segmented, colored mask of (ii) SSL segmentation alone, (iii) SSL segmentation with declumping, **a.iv** and **b.iv** Cellpose segmentation and declumping using "cp_yeastPC_cp3" model ($d = 50$) and "nuclei" model ($d = 30$), respectively.

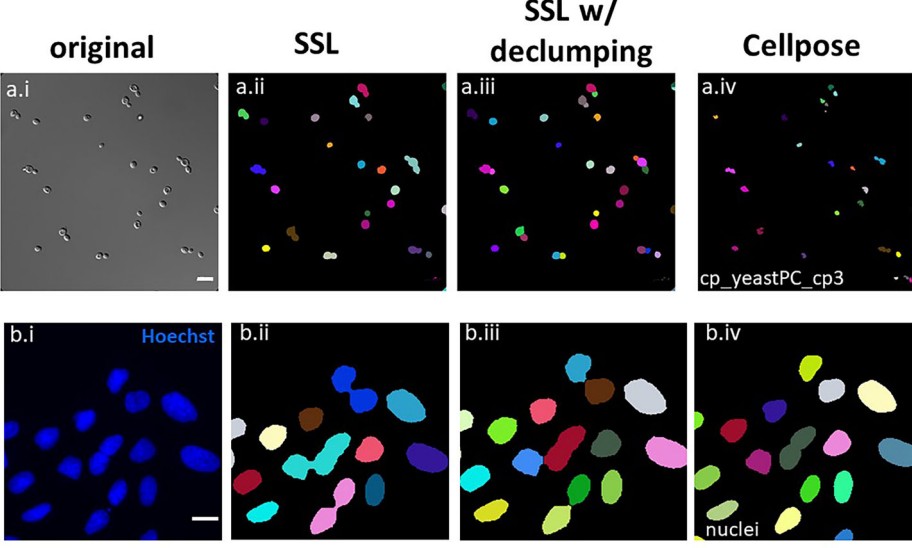

contrast, our SSL approach utilizes transparent and sufficient known data features focusing on entropy, gradient, and intensity without any transformation—each of which can be visualized with regards to "cell" vs "background" contribution. Furthermore, the entire SSL application can be operated on a standard laptop computer, which avoids any extensive hardware installation and data customization for cloud-based uploading. Our innovative SSL method not only shows improvement in segmentation but also reduces the need for advanced computational skills and risk to data security, thereby lowering the barriers for biological researchers integrating computer vision into their work.

Current studies in cell biology often overlook the meticulous analysis of fine cell structures, instead focusing primarily on basic cellular geometry and fluorescence intensity. This could be either attributed to the limitations of robust and reliable segmenting tools, the imaging tools[40] or the data size[41]. The future of cell imaging will need to bridge the gap between high-resolution and high-throughput methodologies. This evolution depends on developing high-throughput cell imaging experiments that are complemented by reliable, straightforward, precise, and robust automated cell segmentation techniques. Here, we have presented results of our SSL algorithm across various modalities and resolutions (Figs. 2 and 8, Table 1) and experimental treatments (Fig. 4) that not only accurately segment cells but also provide detailed structural information (Figs. 4–6) suggesting our code's ability to meet this need.

Future work will continue to expand upon the proof-of-principle declumping algorithms presented here for high cell density imagery of both non-adherent and adherent cell types. The segmentation algorithm we have presented here is foundational to all such subsequent analyses, including declumping, morphological measurements, and the extraction of deep learning parameters—all of which are enhanced by first classifying cells from background with high fidelity. Throughout this study, SSL robustly met this goal across various cell types, cell treatments, and imaging modalities.

## Materials and methods
### Cell culture and microscopy
All mammalian cells were grown in DMEM (ATCC, #30-2002) enriched with 10% fetal bovine serum (ATCC, #30-2020) at 37 °C in a 5% $CO_2$ atmosphere. Cell treatments and growth conditions included a range of environments, including off-the-shelf glass-bottomed dish culturing, or fibronectin-treated surfaces. Detailed culture methodologies and immunofluorescence staining protocols are described in the refs. 17,18,42. Anti-vinculin antibodies (Sigma, V9131) were prepared at a 1:500 concentration. Microscopy information for each cell type, including the mode of microscopy, magnification, numerical aperture, and camera type is provided in Supplementary Note 1. The computer used for code testing was a Dell laptop with CPU: i7-13700H, 13th Gen Intel 14-core, 24MB Cache, 5.0 GHz and GPU: NVIDIA GeForce RTX 4060, 8GB GDDR6; 32GB RAM, DDR5, 4800 MHz.

### Self-supervised learning
The foundational principles of employing OF for SSL-based live cell image analysis were elaborated in earlier publications in which two images from consecutive time points were utilized for labeling and training[17,18]. Here, we broaden the approach to single or fixed-cell imagery, requiring only a single input image for the entire segmentation process (Fig. 1). We extensively investigated a variety of image transformations for simulating motion—including translation, dilation, rotation, and found that Gaussian blurring produced the highest fidelity segmentation results. The success of the Gaussian blur algorithm in enabling OF segmentation was in large part attributed to the fact that both Gaussian blur and OF are conditioned on the constraint that overall image intensity is conserved, thereby resulting in fewer artifacts to be mistakenly labeled by OF as motion. More specific to the Farneback OF algorithm is that blurring mimics the down-sampling utilized by Farneback to create lower resolution images thereby making the two approaches synergistic. A more detailed description of the relationship between Farneback OF and the Gaussian blur filter can be found in Supplementary Note 2.

The Gaussian filter ($\sigma = 0.1$) is applied to the original image to generate a blurred image, which serves as an input for calculating a Farneback OF displacement field. The Farneback algorithm determines OF of a moving object by calculating the displacement field at multiple levels of resolution, starting at the lowest and continuing until convergence.

Cells typically have higher entropy (or texture) than the background due to complex internal structures, such as organelles and the cytoskeleton. We take advantage of this fact by using this entropy difference as a metric for self-tuning and optimizing the OF thresholds that label pixels as either "cell" or "background." This self-tuning phase is termed unsupervised learning as it operates without a predefined training library. Following the unsupervised learning stage, masks for cell and background are created, each being multiplied by the original image. The resultant image products are analyzed to extract features, such as entropy, gradient, and intensity from every labeled pixel (Fig. 1).

The process transitions to supervised learning when feature vectors from each pixel, along with their corresponding labels ("cell" and "background"), are aggregated and introduced to a Naïve Bayes Classifier, employing a ten-fold split for validation. Subsequently, a model is derived

and applied to the original image, culminating in the final segmentation of cells. This methodology underscores a comprehensive approach, integrating both unsupervised and supervised learning phases to achieve precise cell segmentation without the necessity for an extensive training dataset or manual parameter tuning. It is intrinsically adaptive in that a new model is created for each image. The entire process can be implemented using the MATLAB, Python or a graphic-user interface (GUI) supplied as supplementary code packages. A more detailed description of the entire algorithm architecture can be found in Supplementary Note 2.

### Statistics and reproducibility: manual segmentation for F1 score evaluation

Manual segmentation is facilitated through Fiji, utilizing freehand selection tools. The resulting cell masks were then saved and employed to compute the F1 score for SSL, CP, and SSL + CP techniques. The F1 score, a harmonic mean of precision and recall, is defined as

$$F1 = \frac{TP}{TP + \frac{1}{2}(FN + FP)}$$

where TP, FN, and FP are abbreviations for true positive, false negative, and false positive, respectively. This approach enables a quantitative comparison of the segmentation accuracy amongst the three segmenting methods, providing a measure of their performance in terms of both precision (the proportion of true positive results in all positive predictions) and recall (the proportion of true positive results in all actual positives).

### Reporting summary

Further information on research design is available in the Nature Portfolio Reporting Summary linked to this article.

### Data availability

Images evaluated in Figs. 2–4, 6, 8 are available as TIFF files at https://doi.org/10.5281/zenodo.14967876[43].

### Code availability

The SSL application is available in MATLAB, as a stand-alone GUI download for Windows, and as two separate SSL MATLAB source code packages—one with declumping and one without. Additionally, an SSL Python source code without declumping feature is included—one for interface development environment and one for command line interface. All four packages are available for download at https://doi.org/10.5281/zenodo.14967684. They are included here as Supplementary Software 1–4, respectively.

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

## Acknowledgements

V.K.L. gratefully acknowledges support from the National Research Council Research Associateship Program and the Jerome and Isabella Karle Distinguished Scholar Fellowship Program. Funding for this project was provided by the Office of the Undersecretary of Defense through the Tri-Service Biotechnology for a resilient supply chain (T-BRSC) Program and Office of Naval Research through the Naval Research Laboratory's Basic Research Program. In addition, we acknowledge the data provided by the Tissue Microenvironment and Mechanobiology Laboratory of Dr. C.R. at The Catholic University of America.

## Author contributions

V.K.L. Conceptualization, methodology, investigation, data curation, software, visualization, and writing. J.M.B.: conceptualization, methodology, formal analysis, software, visualization, and writing. M.C.R.: conceptualization, methodology, investigation, data curation, software, visualization, and writing. L.K.: methodology, investigation, data curation, software, visualization, and writing. J.A.C.: resources, validation, and writing. M.P.R.: conceptualization, funding acquisition, methodology, investigation, software, visualization, and writing.

## Competing interests

The authors declare no competing interests.
