## [Transparent peer review file · Communications Biology]

A Self-Supervised Learning Approach for High Throughput and High Content Cell Segmentation

Corresponding Author: Dr Marc Raphael

Version 0:

Reviewer comments:

Reviewer #1

(Remarks to the Author)

The major motivations for pushing towards a more generalist, fully automated model with no need for large, labeled datasets or manual interventions are well founded. The proposed SSL approach indeed achieves this across a wide range of imaging modalities and data sets, with a high degree of accuracy as compared to one of the leading models in the field, Cellpose. The underlying methodology to take advantage of optical flows on an augmentation of single images, without the need for time-lapse, is quite novel.

The claim that the software requires little computational expertise is arguable. The code is run in a MATLAB environment, which is perhaps easier to install and set up than a raw python environment, however is not packaged or bundled in a broadly accessible way. Cellpose for instance has many ways of accessing that integrate with other tools in the ecosystem. On top of the native CLI and GUI, it has a ZeroCostDL4Mic implementation, Napari plugin and CellProfiler plugin, among others. The provided code however includes raw source files, and requires the user to install and use MATLAB. Although there is nothing wrong with writing the implementation in MATLAB code, having an end product that is usable without it, perhaps using MATLAB compiler or MATLAB build, would greatly aid in broader access. In addition, providing integrations with existing Bioimage Analysis tools, such as ImageJ or Nepari, would benefit users who are more accustomed to purely graphical interfaces.

In running the code and reproducing the figures, a few issues were encountered. Using a Macbook, with intel architecture, and Matlab 2024, a few alterations to the code were necessary for a successful run. For instance, line 14 of ReadDataCube.m needed to be altered to use Posix style path separators, i.e. "/", rather than Windows style, i.e. "\". Also, the use of a function named "OpticalFlowSegmentation_v2" in "SS_Training_Data.m" needed to be adjusted to simply "OpticalFlowSegmentation", as the former was not defined in "OpticalFlowSegmentation.m".

Reviewer #2

(Remarks to the Author)

This paper presents a method for foreground/background segmentation of cells with a classifier applied per-pixel. The classifier is trained in a self-supervised fashion, based on the optical flow between the image and a slightly blurred version of the image. They demonstrate that their method works on a collection of images of different cell types, imaging modalities, and magnifications. They compare pixel level classification accuracy, via the F1 score, with Cellpose, a convolutional-neural-network-based segmentation algorithm.

The results of the paper are impressive, but there are three major issues:

- Not enough images are used for a proper evaluation compared to existing methods.
- The authors compare to Cellpose, but do not handle "declumping," which Cellpose includes.
- Many details of the algorithm are not clearly explained.

Regarding the first issue: Cellpose 3.0 is now released, and includes references to their downloadable training sets: <https://www.biorxiv.org/content/10.1101/2024.02.10.579780v2.full.pdf#page=12.33> . Using these datasets for evaluation would allow a much larger side-by-side comparison of foreground-background segmentation accuracy. This evaluation should include a side-by-side comparison of computational cost (the timing results in this paper are given without context - is 165 seconds for 15 images fast or slow, compared to alternative approaches? What size images were these?)

The second issue is more qualitative. Declumping (separating individual cells, even when in contact or overlapping) is

critical for analysis of image-based cellular assays. One natural extension of the current paper would be to use the features extracted for the supervised training phase, and use those as a preprocessing for a Cellpose-like network, including training for optical flow for a declumping process. A simpler combination of algorithms would be to use the proposed SSL method for foreground/background segmentation, and the flow vectors predicted by Cellpose for declumping.

The lack of a clear & detailed explanation of the algorithm is a major concern for me. For example, the manuscript says that entropy is used as a ruler for self-tuning. How? Is this detailed in previous papers (I could not locate it in references 17 or 18.)? The method should be (re-)explained here. What features are extracted for the supervised classifier? It appears to be entropy, intensity, and gradient, but the details of their calculation are not apparent. The overall algorithm seems to be built on a fairly straightforward set of foundational methods (optical flow, Naive Bayes classifier, image entropy, etc.) so a detailed explanation of the full algorithm should not be too long or complicated to lay out.

These three are the major points that prevent me from recommending this paper for publication at this time.

Some more technical & lower-level issues:

Use "blurred" instead of "augmented". Augmentation is a more specific term in training image-based classification.

Is the entropy difference between the original and blurred image? Or local entropy (what radius) in the blurred (or original?) image? These details are not explained. (See third issue above.)

Figure 1 is not very clear. "Optical Flow" should be spelled out. The arrows for the Optical Flow result should start from both the original and the blurred image, as they are both required for optical flow. If the entropy calculation depends on both images, then the arrows should communicate that.

The authors refer to "384 and 1536 well plate layouts" multiple times. However, these particular plate layouts are irrelevant. Much more critical is the overall growth in the field, with experiments capturing images for many millions (approaching billions) of cells in experiments, and the more rich staining and imaging modalities that are being applied to these large assays.

The paragraph espousing the "interpretability" of this model is not well-founded. I agree that deep networks are highly uninterpretable, but almost any classifier more complicated than a linear discriminant is also difficult to interpret. Moreover, it's unclear why interpretability is an important concern for a segmentation algorithm.

Signed, Thouis Jones

Reviewer #3

(Remarks to the Author)

Lam et al. present a self supervised algorithm for segmenting cells using optical flow between a blurred version of an image and its original. They demonstrate the algorithm's efficacy on various cell images captured using different modalities. However, the publication of this manuscript is not recommended due to several concerns.

The primary reason for rejecting the manuscript is lack novelty. The authors previously developed and published their method for live cell imaging and this manuscript does not offer significant advancement. The authors have simply replaced a time series of two images with a blurred and an original version of single time point.

Another substantial concern is the method's limited performance when dealing with touching cells. The vast majority of high content screens employ dense seeding or 3D cell culture such as organoids, tumoroids and other microphysiological systems. In Sup Fig 2 the declumping yield poor segmentation results. Moreover, additional masks appear after declumping and the methods used for this step have not been described. SSL performs instance segmentation instead of semantic segmentation. Therefore, introducing a declumping step is essential, and such a successful and well-documented advancement could have constituted sufficient novelty for publication.

A further obstacle to the publication of this manuscript is the implementation of the algorithm. The use of proprietary software for analyzing terabytes of imaging data is problematic. Most Pharma companies involved in HCS use either an in-house cluster or the cloud for analysing their data. When implementing their own algorithms, they generally use python. I will illustrate the rationale with a recent example from my lab. We recently carried out a screen that generated 74TB of data (0.5 mio images/plate, 2 TB/plate). Parallelisation is a must to analyse such a dataset. The authors cite a time of 165s for analysing 15 images. This would translate to 140h (6 days) for a single plate of the above mentioned screen. The authors did not state whether that time included a declumping step or feature extraction step. The authors also fail to mention the size of the images (MB).

Lastly, upon attempting to test their SSL algorithm on my Mac operating system, I encountered a compatibility issue, as the software is hardcoded for Windows.

Minor points.

- The authors suggest that the 384-well format and high-resolution immersion lens imaging are recent developments. However, these technologies have been available for decades.

- High content screening studies typically involve large sample sizes, and there is no shortage of large imaging datasets accessible on the internet. When proposing a method for large-scale applications, utilizing one of the massive datasets available online is essential. Problems often become apparent only when algorithms are applied at scale, making such a test crucial.

- For the supervised learning, the authors use Naive Bayesian Classifier. Did they try other classifiers?

- fully connected FPN should be briefly described in the introduction.

- The term "unstainable" should be replaced with "unsustainable" in the sentence "In addition, the increasing requirement of extracting fine cellular details from this imagery means that curated training data sets will need to grow exponentially at a rate that, in our view, is simply unstainable."

Version 1:

Reviewer comments:

Reviewer #1

(Remarks to the Author)

The inclusion of a GUI using the freely available MATLAB runtime drastically improves accessibility. I applaud the authors for the inclusion.

The port to python is nice, in that the python interpreter is freely available, however should at the very least provide a simple argparse CLI entrypoint for defining the directory path, rather than having to edit a hardcoded path in the demo script. It would also be beneficial to include a basic PEP-compliant pyproject.toml to make the library into a proper package, and also list the dependencies (as opposed to the blank, comment-only requirements.txt currently provided). These are very straight forward additions, which are implementable quickly.

I agree with the authors that integrating with existing tools, via plugins, is beyond the scope of the manuscript, and highly encourage them to follow through in the future in providing this functionality. Alternatively, there are a number of open-source tools to very quickly spin up a basic GUI for a python codebase. Gradio and DearPyGui come to mind, but there are countless others. Providing a no-code option to readers of the manuscript dramatically increases the likelihood that they will use the developed algorithms.

Reviewer #2

(Remarks to the Author)

Thank you to the authors for addressing my previous comments. I am happy to recommend publications.

Reviewer #3

(Remarks to the Author)

I am impressed by the efforts the authors have undertaken to address the points I raised earlier. The manuscript is now much more informative and the python implementation allows for broader accessibility, although a yaml file should be added to create a compatible environment.

I have successfully run the python code on a dataset of a z-stack of organoid nuclei at 10x with each image 94MB in size (10000x10000 px). The code ran very slowly, but could be further improved by the community using dask arrays for instance. The result of the segmentation were disappointing as the algorithm missed many objects and the masks were all missing parts of the organoids (see attached screen shots). Nevertheless, I find it is an encouraging starting point and would warrant some efforts towards optimising the accuracy. Maybe there are some parameters that still could be tuned by the user depending on resolution and dynamic range of the image?

I believe that with modifications the current version of the manuscript is acceptable for publication. I believe the community will profit of this approach.

Reviewer #1 (Remarks to the Author):

The major motivations for pushing towards a more generalist, fully automated model with no need for large, labeled datasets or manual interventions are well founded. The proposed SSL approach indeed achieves this across a wide range of imaging modalities and data sets, with a high degree of accuracy as compared to one of the leading models in the field, Cellpose. The underlying methodology to take advantage of optical flows on an augmentation of single images, without the need for time-lapse, is quite novel.

The claim that the software requires little computational expertise is arguable. The code is run in a MATLAB environment, which is perhaps easier to install and set up than a raw python environment, however is not packaged or bundled in a broadly accessible way. Cellpose for instance has many ways of accessing that integrate with other tools in the ecosystem. On top of the native CLI and GUI, it has a ZeroCostDL4Mic implementation, Napari plugin and CellProfiler plugin, among others. The provided code however includes raw source files, and requires the user to install and use MATLAB. Although there is nothing wrong with writing the implementation in MATLAB code, having an end product that is usable without it, perhaps using MATLAB compiler or MATLAB build, would greatly aid in broader access. In addition, providing integrations with existing Bioimage Analysis tools, such as ImageJ or Nepari, would benefit users who are more accustomed to purely graphical interfaces.

We appreciate the Reviewer pointing out the need for more accessible forms of the code. In response, we provide two more options for running the code.

1. To address the Reviewer's point regarding easier integration with other tools in the ecosystem, we have created a python version of the code. While making a plug-in for the other software packages mentioned is beyond the scope of the current manuscript, having the code in Python is an important first step for integration with other commonly utilized image analysis tools such as CellPose, Napari and CellProfiler. This new python code package has been uploaded to the Manuscript Central website along with a README document and will be made publicly available pending manuscript acceptance. The code is designed to be used with tiff images for the automated segmentation of cells from background. This code was tested on Python 3.11.5 64-bit | Qt 5.15.2 | PyQt5 5.15.7 | Windows 10 using Spyder IDE 5.4.3 distributed by Anaconda.
2. Again taking the Reviewer's advice, we have created a streamlined GUI that can be run in the MATLAB Runtime environment. As shown in the screenshot below, the GUI allows the user to load images from the manuscript or their own data. After selecting the images, the only step is to hit the "PROCESS" button, thereby retaining the self-supervised learning goal of being free from requiring curated training sets or parameter tuning. We note that because MATLAB Runtime is free, we believe this implementation also widens the accessibility of the code. This GUI has been uploaded to the Manuscript Central website along with a README document describing how to install the MATLAB Runtime environment and will be made publicly available pending manuscript acceptance.

Our Mathworks licensing agreement only allows us to generate the GUI as a Windows executable, but for future releases we plan a Mac compatible GUI as well.

In running the code and reproducing the figures, a few issues were encountered. Using a Macbook, with intel architecture, and Matlab 2024, a few alterations to the code were necessary for a successful run. For instance, line 14 of ReadDataCube.m needed to be altered to use Posix style path separators, i.e. "/", rather than Windows style, i.e. "\". Also, the use of a function named "OpticalFlowSegmentation_v2" in "SS_Training_Data.m" needed to be adjusted to simply "OpticalFlowSegmentation", as the former was not defined in "OpticalFlowSegmentation.m".

We thank the Reviewer for catching these oversights. We have updated the naming of the OpticalFlowSegmentation function and included a note regarding the Posix style path separators.

Reviewer #2 (Remarks to the Author):

This paper presents a method for foreground/background segmentation of cells with a classifier applied per-pixel. The classifier is trained in a self-supervised fashion, based on the optical flow between the image and a slightly blurred version of the image. They demonstrate that their method works on a collection of images of different cell types, imaging modalities, and magnifications. They compare pixel level classification accuracy, via the F1 score, with Cellpose, a convolutional-neural-network-based segmentation algorithm.

The results of the paper are impressive, but there are three major issues:

- Not enough images are used for a proper evaluation compared to existing methods.*
- The authors compare to Cellpose, but do not handle "declumping," which Cellpose includes.*

Regarding the first issue: Cellpose 3.0 is now released, and includes references to their downloadable training sets: <https://www.biorxiv.org/content/10.1101/2024.02.10.579780v2.full.pdf#page=12.33> . Using these datasets for evaluation would allow a much larger side-by-side comparison of foreground-background segmentation accuracy. This evaluation should include a side-by-side comparison of computational cost (the timing results in this paper are given without context - is 165 seconds for 15 images fast or slow, compared to alternative approaches? What size images were these?)

We appreciate this input. To address the Reviewer's request for larger datasets we broadened the number images analyzed to over 800 images and then compared the SSL results to those of Cellpose in a new table and a new figure with representative imagery. The revised manuscript now includes a detailed explanation of this larger dataset analysis which we reprint here. In addition, we have further delineated table in the SI to emphasize that finding the optimized Cellpose model required the user to test various Cellpose models and diameters. That SI table is also reprinted below:

Quantitative analysis of SSL and Cellpose performance on larger datasets

To broaden these results, a comparison between SSL and Cellpose (CP) models was performed to evaluate segmentation performance across various biological objects, microscope modalities, image sizes and fluorescent labels using hundreds of in-house and public data set images. Table 1 details the resulting F1 scores, processing times, and – because Cellpose sometimes failed to segment entire images – the percentage of images successfully segmented. For the 10X epifluorescence and phase contrast Hs27 in-house datasets, the SSL F1 scores were comparable to Cellpose cyto2, but with significantly faster processing times, saving 25 minutes across over 500 images on the same CPU. For the 40X Hs27 images, SSL achieved a higher F1 score of 0.831 and a faster processing time compared to Cellpose cyto2, which also failed to segment approximately 12% of the images. A similar trend was observed for the 60X Hs27 images stained for F-actin and vinculin in which the cyto2 and cyto3 F1 scores were 0.608 and 0.454, respectively, in contrast to a significantly higher F1 score of 0.758 for SSL. In addition, the Cellpose models required more processing time while successfully segmenting only about half those images.

When evaluating a Cell Paint dataset of Hoechst-stained nuclei that was also used to train the Cellpose 'nuclei' model, Cellpose outperformed SSL, achieving a F1 score of 0.953 with 100% segmented in 58 minutes, versus a 0.873 SSL F1 score in 23.5 minutes. In general, the table shows that SSL demonstrates faster processing times and higher fidelity segmentation, particularly for the in-house Hs27 10X, 40X and

60X datasets, compared to Cellpose. Furthermore, as described above, SSL does not require that manual tuning of a diameter parameter for optimization.

Figure 8 presents representative image segmentations from Table 1 by SSL (Figures 8a, c, e) and Cellpose (Figures 8b, d and f). In most scenarios it is clear by eye that Cellpose has a tendency towards false negatives which lower its respective F1 scores in Table 1. The exception is with the nuclei segmentation, where SSL tended towards over-segmentation (Figure 8e) compared to the 'nuclei' model of Cellpose (Figure 8f). Additionally, SSL counted surrounding flow signals near neighboring nuclei, reducing segmentation precision relative to Cellpose. However, as noted above, Cellpose was specifically trained on these images while still requiring the manual tuning of a diameter ($d = 30$) whereas SSL is self-trained and completely automated.

	Epifluorescent Hs27 10X		Phase Contrast Hs27 10X		Epifluorescence Hs27 40X		Epifluorescence Hs27 60X			Epifluorescence Nucleus	
Data source	in-house		in-house		in-house		in-house			Cell Painting	
Stains	F-actin, Vinculin		N/A		Vinculin		F-actin, Vinculin			Hoechst 33342	
No. of images	20		565		17		67			189	
Image Size	2304 × 2304		1344 × 1024		1324 × 1024		2262 × 2262			696 × 520	
Segmentation technique	SSL	Cellpose	SSL	Cellpose	SSL	Cellpose	SSL	Cellpose		SSL	Cellpose
Model *	-	cyto2	-	cyto2	-	cyto2	-	Cyto2	cyto3	-	nuclei
Diameter	-	100	-	100	-	150	-	400	400	-	30
F1	0.888	0.882	0.771	0.777	0.831	0.818	0.758	0.608	0.454	0.873	0.953
Times (mins)	10.48	16.67	123	148.05	3.02	5.15	19.04	33.78	40.53	23.45	58.00
% of exported segmentation images	100%	100%	100%	100%	100%	88.23%	100%	62.69%	53.73%	100%	100%

Table 1: Segmentation performance by SSL and an optimized Cellpose model for a range of imaging datasets. The table details information on data source, optical modalities, image size and required Cellpose inputs. Output metrics include F1 scores, processing times, and percentage of images segmented from the data set. d: diameter (in pixels), *Only required by Cellpose.

Figure 8: Representative image segmentation results from Table 1 by SSL and an optimized Cellpose model, (a) and (b) 10X Hs27 stained with Vinculin, (c) and (d) 60X Hs27 stained with F-actin and Vinculin, (e) and (f) nucleus stained with Hoechst. Scale bar: a-b: 50 μ m, c-d: 25 μ m, e-f: 20 μ m.

Supplemental Note 3: Summary of segmentation performance by SSL and different Cellpose models on several imaging datasets.

	Model	% of exported segmentation images	F1 score	Time (minutes)
Epifluorescence Hs27 60X Number of images = 67	SSL	100%	0.758	19.03
	cp_cyto (d=50)	82.09%	0.078	12.07
	cp_cyto (d=100)	49.25%	0.119	8.33
	cp_cyto (d=200)	38.81%	0.192	9.47
	cp_cyto (d=300)	29.85%	0.253	8.11
	cp_cyto (d=400)	35.82%	0.339	13.73
	cp_cyto2 (d=50)	100%	0.059	12.54
	cp_cyto2 (d=100)	85.07%	0.195	10.58
	cp_cyto2 (d=200)	82.09%	0.455	15.64
	cp_cyto2 (d=300)	68.66%	0.674	26.83
	cp_cyto2 (d=400)	62.69%	0.608	33.80
	cp_cyto3 (d=50)	86.57%	0.072	14.78
	cp_cyto3 (d=100)	70.15%	0.191	17.21
	cp_cyto3 (d=200)	74.63%	0.337	36.2
	cp_cyto3 (d=300)	67.16%	0.400	43.95
cp_cyto3 (d=400)	53.73%	0.454	38.63	
Epifluorescence Hs27 10X Number of images = 20	SSL	100%	0.8877	10.49
	cp_cyto2 (d = 50)	100%	0.795	11.05
	cp_cyto2 (d = 100)	100%	0.882	16.67
	cp_cyto2 (d = 150)	100%	0.792	16.56
Phase Contrast Hs27 10X Number of images = 565	SSL	100%	0.771	129.33
	cp_cyto2 (d = 30)	100%	0.622	142.00
	cp_cyto2 (d = 50)	100%	0.777	148.05
	cp_cyto2 (d = 100)	100%	0.614	118.7
Epifluorescence Nucleus Number of images = 189	SSL	100%	0.873	23.47
	cp_nuclei (d = 30)	100%	0.953	58.01
	SSL + cp_nuclei (d=30)	100%	0.944	76.03
	SSL + declumping	100%	0.873	24.22
Epifluorescence Hs27 40X Number of images = 17	SSL	100%	0.831	3.02
	cp_cyto2 (d=150)	82.35%	0.818	5.154
DIC S. cerevisiae 63X Number of images = 60	SSL	100%	N/A	36.35
	yeast_PhC_cp3 (d= 50)	100%	N/A	65.67
	SSL + declumping	100%	N/A	39.14

The second issue is more qualitative. Declumping (separating individual cells, even when in contact or overlapping) is critical for analysis of image-based cellular assays. One natural extension of the current paper would be to use the features extracted for the supervised training phase, and use those as a preprocessing for a Cellpose-like network, including training for optical flow for a declumping process. A simpler combination of algorithms would be to use the proposed SSL method for foreground/background segmentation, and the flow vectors predicted by Cellpose for declumping.

While this work primarily focuses on applications to cell confluency and intracellular feature analysis - which do not require declumping - we agree that declumping algorithms are very useful and would make an important addition to the current code base. We did start work on the declumping problem for this publication but upon realizing that it was a whole project unto itself, decided to continue to work on it separately.

During our initial evaluation of declumping software we tested both Cellpose as well as the newest Sartorius Incucyte CNN approach trained on the 1.7 million cell LIVECell dataset published in Nature Methods (www.nature.com/articles/s41592-021-01249-6). The results varied widely across the diverse images used in this publication. For instance, below is a result from the \$15K Incucyte software on one of our Hs27 fibroblast images in which it both failed to segment whole cells and then poorly declumped the cells that were segmented. Thus, we believe a general solution to declumping is still very much a work in progress, even for the what are considered to be the best solutions on the market.

That said, we like the Reviewer's idea of using SSL segmented cell output as input into additional declumping algorithms and so have added new results to the manuscript related to that approach. In particular, for non-adherent cells and cell nuclei we found that inputting our SSL segmented cell clumps into the watershed algorithm gave promising preliminary results (see newly added Figure 9 below).

So far, we have not had as much success with adherent cells, which we attempted to declump by using the SSL segmented cell clumps as inputs into Cellpose. In general, Cellpose did as well or better

analyzing the raw images than when working with SSL segmented output, but we still have much to explore with regards to this approach.

In response to the Reviewer's comments, we have emphasized in numerous places in the revised manuscript that while not the goal of this particular work, we feel declumping is an important future addition and have begun work on the problem as indicated by the new Figure 9 and explanatory text:

While this work is focused on automated segmentation for quantifying cell confluency and detailed intracellular features, we recognize that identifying individual cells within clumps is a naturally desirable extension of the current algorithm. Here we demonstrate that the application of the watershed algorithm to SSL segmented clumps enables the declumping of both *S. cerevisiae* and nuclei. We view this as proof of principle that SSL segmentation results can be utilized as downstream inputs for additional processing applications such as declumping. Figure 9 shows representative imagery and declumping results for both SSL and Cellpose (cp_yeastPC_cp3 model) using our in-house 63X DIC images of *S. cerevisiae* (n = 60); and for the nuclei images of Table 1. SSL with watershed declumping performed especially well on the yeast imagery compared to Cellpose which often failed to segment cells in the clump.

This new code package has been uploaded as a separate Matlab package and will continue to develop this feature for both non-adherent and adherent cell types in future work.

Figure 9: Representative segmentation results by SSL and the best Cellpose model on (a) 63X *S. cerevisiae*, and (b) nucleus stained with Hoechst. From left to right, (i) original image of image to-be-segmented, colored mask of (ii) SSL segmentation alone, (iii) SSL segmentation with add-on declumping, (a.iv) and (b.iv) Cellpose segmentation alone using 'cp_yeastPC_cp3' model (d = 50) and 'nuclei' model (d=30), respectively.

The lack of a clear & detailed explanation of the algorithm is a major concern for me. For example, the manuscript says that entropy is used as a ruler for self-tuning. How? Is this detailed in previous papers (I could not locate it in references 17 or 18.)? The method should be (re-)explained here. What features are extracted for the supervised classifier? It appears to be entropy, intensity, and gradient, but the details of their calculation are not apparent. The overall algorithm seems to be built on a fairly straightforward set of foundational methods (optical flow, Naive Bayes classifier, image entropy, etc.) so a detailed explanation of the full algorithm should not be too long or complicated to lay out.

We agree with the Reviewer that a more detailed explanation of the algorithm is warranted. The following description is somewhat lengthy and so will be added to the Supporting Information section of the revised manuscript:

We describe our self-supervised machine learning algorithm in the context of the three primary machine learning steps: Label, Train and Test.

Unsupervised Labeling

We previously demonstrated that the Farneback optical flow (OF) algorithm is a useful tool for self-supervised live cell segmentation due to the fact that motion is intrinsic throughout live cell microscopy. From membrane fluctuations at the cell's perimeter to the intracellular fluctuations of organelles, vesicles and granules, motion is ubiquitously observed in live cells, enabling robust segmentation to be accomplished via Farneback OF.

Here we show that the same approach can work with fixed cells or single frame imagery *if that motion is simulated in manner conducive to the optical flow algorithm*. We extensively investigated a variety of image transformations for simulating motion – including translation, dilation, rotation (data not shown) - and found that Gaussian blurring produced the highest fidelity segmentation results. The success of the Gaussian blur algorithm in enabling OF segmentation was in large part attributed to the fact that both Gaussian blur and OF are conditioned on the constraint that overall image intensity is conserved, thereby resulting in fewer artifacts to be mistakenly labeled by OF as motion. More specific to the Farneback OF algorithm is that blurring mimics the down-sampling utilized by Farneback to create lower resolution images thereby making the two approaches synergistic. The two images are represented at top level of Figure A in the orange rectangles. To save the Reviewer time we have also reprinted below our description of how Farneback OF utilizes a multi-resolution approach for enabling robustness to noise and image jitter.

The second level of Figure A is shown in blue to indicate an optical flow 'image' in which each pixel value is the flow magnitude returned by the Farneback algorithm. Some of those flow magnitudes will be below a background threshold and labeled 'background' (BkgMag), some will be above a cell threshold and labeled 'cell' (CellMag), while others remain unlabeled until the testing step. To ensure large enough training data sets, the OF threshold is incremented as indicated by the arrows on the blue rectangles at the third level down in Figure A. While sufficiently large data sets could be determined by simply summing the number of pixels with 'cell' and 'background' labels, we found the algorithm to be more robust if we summed over the entropy associated with each pixel (7x7 pixel neighborhood) since pixel entropy was incorporated into the supervised training step that follows.

Figure A. Unsupervised labeling of pixels identified by Farneback optical flow as either ‘background’ or ‘cell’. ‘th’ are thresholds for cell and background labeling via optical flow magnitude values. Unlabeled pixels are segmented in the testing step.

Supervised Training and Testing of Unlabeled Pixels

Training of the Naïve Bayes Classifier is accomplished via feature extraction from the labeled pixels on the blurred image. In this work single pixel-based training from the following three features proved robust: entropy (7x7 pixel neighborhood), image gradient (5x5 pixel neighborhood, method ‘intermediate’) and pixel intensity. This is indicated in Figure B with labeled pixels indicated by the orange rectangles and the extracted features indicated by the blue rectangles. The feature vectors are used to train a Naïve Bayes classifier model which in turn is used to test previously unlabeled pixels on the original image.

In constructing our algorithm, we initially explored classifiers such as random forests, SVM, and K Nearest Neighbor. However, the Naïve Bayes classifier was chosen as a flexible and effective option, as they are known to have good bias-variance tradeoff because of their simplistic assumption of feature independence, and was found to perform robustly in the context of cell segmentation outlined here.

Figure B. Supervised training via entropy, gradient and intensity features extracted from the labelled pixels. Unlabelled pixels are subsequently tested and labelled based on the resulting Naïve Bayes classifier model.

Fitting Surfaces To Images To Find The Displacement Field (Previously Published in the SI of ‘A Self-Supervised Learning Approach for High Throughput and High Content Cell Segmentation’ <https://www.nature.com/articles/s42003-022-04117-x>)

We can think of a grayscale image, I , as a cloud of points in three dimensions where the intensity of the pixel is the height, $z = I(i, j)$, above the corresponding locations (i, j) in the image plane with coordinates x and y (Fig S2). A natural idea is to fit a smooth function, $F(x, y)$, to the discrete samples represented by the image, $I(i, j)$. Unlike the image itself, the function is defined at every location (x, y) in the plane of the image not just the locations of the pixels (i, j) . Also, the function has a continuum of intensity values not just at the discrete values of the image format (e.g, the integers from 0-255 for 8-bit images). Now, consider fitting a function to each of a pair of consecutive images in time, I_1 and I_2 . The central premise of optical flow is that the corresponding functions, F_1 and F_2 , can be related to each other by a spatial shifting of pixel values, called a displacement field, $\mathbf{d}(x, y)$: $F_2 = F_1(\mathbf{x} - \mathbf{d})$ where $\mathbf{x} = (x, y)$.

Fig S2. An example using a three-level image pyramid to compute the displacement field, $\mathbf{d}(x, y)$. The original image pair, I_1 and I_2 , is progressively down-sampled to create lower-resolution versions. Image surface functions, f , are fit to local patches in the images at each level of the pyramid. Beginning with the lowest resolution functions, f_1^{low} , and f_2^{low} , a displacement field, \mathbf{d}_{low} , is calculated and, then, up-sampled, $\tilde{\mathbf{d}}_{\text{low}}$, to the resolution of the middle layer images in the pyramid. This process is repeated for the middle layer to provide an initial displacement field, $\tilde{\mathbf{d}}_{\text{mid}}$, for the determining the final displacement field, \mathbf{d} , between the original images.

Optical flow methods differ on how this displacement field is computed. Global methods (e.g., Horn-Schunck algorithm) solve for the displacement field across the entire image simultaneously and iterate this process to achieve a prescribed level of smoothness. While sensitive to noise, this approach can create a dense displacement field even in the interior of constant brightness regions. Local methods (e.g., Lucas-Kanade algorithm) solve for the displacement field in local neighborhoods of pixels or patches using linear regression, typically. This approach is more robust to noise but requires very small differences between the local patches being compared between images and is effective at high image acquisition rates compared to motion in the images. Another class of optical flow methods (e.g. Farneback algorithm) operate between the global and local forms via a multi-resolution approach that iteratively determines the displacement field between a series of lower resolution versions of the original images.

The multi-resolution approach combines the robustness to noise and global image shifts while being able to track the deformation of cells occurring at varying length scales and microscope magnifications. We have chosen this optical flow method for the automatic semantic segmentation module of our self-supervised learning approach. The method we use corresponds to fitting a 2nd-order polynomial in x and y to a local patch of the image and then repeating this process across the entire image to approximate the surface functions. As already described for local methods, this patch-wise reshaping of the image

surface f_1 into f_2 , at a particular resolution is more effective if the shifts are not large compared to the size of neighborhood used to fit the polynomials in each image.

Some more technical & lower-level issues:

Use "blurred" instead of "augmented". Augmentation is a more specific term in training image-based classification.

Corrected. Thank you.

Is the entropy difference between the original and blurred image? Or local entropy (what radius) in the blurred (or original?) image? These details are not explained. (See third issue above.)

Described in algorithmic description above.

Figure 1 is not very clear. "Optical Flow" should be spelled out. The arrows for the Optical Flow result should start from both the original and the blurred image, as they are both required for optical flow. If the entropy calculation depends on both images, then the arrows should communicate that.

This figure has been updated in the revised manuscript.

The authors refer to "384 and 1536 well plate layouts" multiple times. However, these particular plate layouts are irrelevant. Much more critical is the overall growth in the field, with experiments capturing images for many millions (approaching billions) of cells in experiments, and the more rich staining and imaging modalities that are being applied to these large assays.

We agree with the Reviewer that our wording here could have been more clear. Our point is that more wells enable vastly more experimental conditions per well plate. This in turn enables a wider spectrum of cell phenotypes that require high fidelity segmentation (e.g. amoeboid, mesenchymal, epithelial). Our SSL approach is well suited for this scenario because the model is retrained on every image rather than relying on a static library applied to all images. We have updated the manuscript accordingly.

The paragraph espousing the "interpretability" of this model is not well-founded. I agree that deep networks are highly uninterpretable, but almost any classifier more complicated than a linear discriminant is also difficult to interpret. Moreover, it's unclear why interpretability is an important concern for a segmentation algorithm.

We agree that 'interpretability' is not the most helpful term for this scenario and have replaced it with the term 'traceability' in the manuscript. By this we mean the ability to *trace backwards* from the segmentation results to the image feature vectors that proved most influential.

As a representative example, below we compare 'cell' and 'background' feature histograms for entropy (top) and gradient (bottom). The clear separation in entropy histograms is an indicator that entropy

methods (and related features) are worthy of further exploration. In contrast, the significant overlap in the gradient histograms indicates that it is not as dominant a feature for segmentation – a fact we can confirm by removing it from the training.

(Although reproduced from the SI of ‘A Self-Supervised Learning Approach for High Throughput and High Content Cell Segmentation’ <https://www.nature.com/articles/s42003-022-04117-x> this histogram figure is typical of results in this manuscript as well.)

We find the ability to trace segmentation results back to the dominant feature vectors in this manner a helpful tool in directing future code development.

Reviewer #3 (Remarks to the Author):

Lam et al. present a self supervised algorithm for segmenting cells using optical flow between a blurred version of an image and its original. They demonstrate the algorithm's efficacy on various cell images captured using different modalities. However, the publication of this manuscript is not recommended due to several concerns.

The primary reason for rejecting the manuscript is lack novelty. The authors previously developed and published their method for live cell imaging and this manuscript does not offer significant advancement. The authors have simply replaced a time series of two images with a blurred and an original version of single time point.

We understand the Reviewer's point that we could have done a better job explaining the work that went into this advance and why it was not so simple to discover. We have updated the manuscript and SI with additional context in this regard although we still feel that to include such data would be a distraction from the goals of this paper which is to emphasize the applicability of SSL to both high throughput and high content imaging.

From the revised manuscript:

We extensively investigated a variety of image transformations for simulating motion – including translation, dilation, rotation (data not shown) - and found that Gaussian blurring produced the highest fidelity segmentation results. The success of the Gaussian blur algorithm in enabling OF segmentation was in large part attributed to the fact that both Gaussian blur and OF are conditioned on the constraint that overall image intensity is conserved, thereby resulting in fewer artifacts to be mistakenly labeled by OF as motion. More specific to the Farneback OF algorithm is that blurring mimics the down-sampling utilized by Farneback to create lower resolution images thereby making the two approaches synergistic. A more detailed description of the relationship between Farneback optical flow and the Gaussian blur filter can be found in Supplementary Notes.

Another substantial concern is the method's limited performance when dealing with touching cells. The vast majority of high content screens employ dense seeding or 3D cell culture such as organoids, tumoroids and other microphysiological systems. In Sup Fig 2 the declumping yield poor segmentation results. Moreover, additional masks appear after declumping and the methods used for this step have not been described. SSL performs instance segmentation instead of semantic segmentation. Therefore, introducing a declumping step is essential, and such a successful and well-documented advancement could have constituted sufficient novelty for publication.

While this work primarily focuses on applications to cell confluency and intracellular feature analysis - which do not require declumping - we agree that declumping algorithms are very useful and would make an important addition to the current code base. We did start work on the declumping problem for this publication but upon realizing that it was a whole project unto itself, decided to continue work on it separately.

During our initial evaluation of declumping software we tested both Cellpose as well as the new Incucyte CNN approach trained on the 1.7 million cell LIVECell dataset (www.nature.com/articles/s41592-021-

01249-6). As you can imagine, the results varied widely across the diverse images used in this publication.

We have begun work using SSL segmented cell output as input into additional declumping algorithms and so have added new results to the manuscript related to that approach. In particular, for non-adherent cells and cell nuclei we found that inputting our SSL segmented cell clumps into the watershed algorithm gave promising preliminary results (see newly added Figure 9 below).

So far, we have not had as much success with adherent cells, which we attempted to declump by using the SSL segmented cell clumps as inputs into the Cellpose. In general, Cellpose did as well or better analyzing the raw images than when working with SSL segmented output, but we still have much to explore with regards to this approach.

In response to the Reviewer's comments, we have emphasized in numerous places in the revised manuscript that while not the goal of this particular work, we feel declumping is an important future addition and have begun work on the problem as indicated by the new Figure 9 and explanatory text:

While this work is focused on automated segmentation for quantifying cell confluency and detailed intracellular features, we recognize that identifying individual cells within clumps is a naturally desirable extension of the current algorithm. Here we demonstrate that the application of the watershed algorithm to SSL segmented clumps enables the declumping of both *S. cerevisiae* and nuclei. We view this as proof of principle that SSL segmentation results can be utilized as downstream inputs for additional processing applications such as declumping. Figure 9 shows representative imagery and declumping results for both SSL and Cellpose (cp_yeastPC_cp3 model) using our in-house 63X DIC images of *S. cerevisiae* (n = 60); and for the nuclei images of Table 1. SSL with watershed declumping performed especially well on the yeast imagery compared to Cellpose which often failed to segment cells in the clump. We will continue to develop this feature for both non-adherent and adherent cell types in future work.

Figure 9: Representative segmentation results by SSL and the best Cellpose model on (a) 63X *S. cerevisiae*, and (b) nucleus stained with Hoestch. From left to right, (i) original image of image to-be-segmented, colored mask of (ii) SSL segmentation alone, (iii) SSL segmentation with add-on declumping, (a.iv) and (b.iv) Cellpose segmentation alone using ‘cp_yeastPC_cp3’ model (d = 50) and ‘nuclei’ model (d=30), respectively.

A further obstacle to the publication of this manuscript is the implementation of the algorithm. The use of proprietary software for analyzing terabytes of imaging data is problematic. Most Pharma companies involved in HCS use either an in-house cluster or the cloud for analysing their data. When implementing their own algorithms, they generally use python. I will illustrate the rationale with a recent example from my lab. We recently carried out a screen that generated 74TB of data (0.5 mio images/plate, 2 TB/plate). Parallelisation is a must to analyse such a dataset. The authors cite a time of 165s for analysing 15 images. This would translate to 140h (6 days) for a single plate of the above mentioned screen. The authors did not state whether that time included a declumping step or feature extraction step. The authors also fail to mention the size of the images (MB). Lastly, upon attempting to test their SSL algorithm on my Mac operating system, I encountered a compatibility issue, as the software is hardcoded for Windows.

We appreciate the Reviewer pointing out the need for more accessible forms of the code. In response, we provide two more options for running the code.

1. To address the Reviewer’s point regarding easier integration with other tools in the ecosystem, we have created a python version of the code. While making a plug-in for the other software packages mentioned is beyond the scope of the current manuscript, having the code in Python is an important first step for integration with other commonly utilized image analysis tools such as CellPose, Napari and CellProfiler. This new python code package has been uploaded to the Manuscript Central website along with a README document and will be made publicly available pending manuscript acceptance. The code is designed to be used with tiff images for the

automated segmentation of cells from background. This code was tested on Python 3.11.5 64-bit | Qt 5.15.2 | PyQt5 5.15.7 | Windows 10 using Spyder IDE 5.4.3 distributed by Anaconda.

2. We have also created a streamlined GUI that can be run in the MATLAB Runtime environment. As shown in the screenshot below, the GUI allows the user to load images from the manuscript or their own data. After selecting the images, the only step is to hit the "PROCESS" button, thereby retaining the self-supervised learning goal of being free from requiring curated training sets or parameter tuning. We note that because MATLAB Runtime is free, we believe this implementation also widens the accessibility of the code. This GUI has been uploaded to the Manuscript Central website along with a README document describing how to install the MATLAB Runtime environment and will be made publicly available pending manuscript acceptance.

Our Mathworks licensing agreement only allows us to generate the GUI as a Windows executable, but for future releases we plan a Mac compatible GUI as well.

The reviewer is correct that in using a Mac a few alterations to the code are necessary for a successful run. In particular, line 14 of ReadDataCube.m needs to be altered to use Posix style path separators, i.e. "/", rather than Windows style, i.e. "\".

We have included a note regarding the Posix style path separators.

Minor points.

- The authors suggest that the 384-well format and high-resolution immersion lens imaging are recent developments. However, these technologies have been available for decades.

We agree with the Reviewer that our wording here could have been more clear. Our point is that more wells and higher resolutions enable vastly more experimental conditions and features to be extracted per well plate, respectively. This in turn enables a wider spectrum of cell phenotypes that require high fidelity segmentation (e.g. amoeboid, mesenchymal, epithelial). Our SSL approach is well suited for this scenario because the model is retrained on every image rather than relying on a static library applied to all images. We have updated the manuscript accordingly.

- High content screening studies typically involve large sample sizes, and there is no shortage of large imaging datasets accessible on the internet. When proposing a method for large-scale applications, utilizing one of the massive datasets available online is essential. Problems often become apparent only when algorithms are applied at scale, making such a test crucial.

To address the Reviewer's request for larger datasets we broadened the number images analyzed to over 800 images and then compared the SSL results to those of Cellpose in a new table and a new figure with representative imagery. The revised manuscript now includes a detailed explanation of this larger dataset analysis which we reprint here. In addition, we have further delineated the new manuscript table in the SI to emphasize that finding the optimized Cellpose model required the user to test various Cellpose models and diameters. That SI table is also reprinted below:

Quantitative analysis of SSL and Cellpose performance on larger datasets

To broaden these results, a comparison between SSL and Cellpose (CP) models was performed to evaluate segmentation performance across various biological objects, microscope modalities, image sizes and fluorescent labels using hundreds of in-house and public data set images. Table 1 details the resulting F1 scores, processing times, and – because Cellpose sometimes failed to segment entire images – the percentage of images successfully segmented. For the 10X epifluorescence and phase contrast Hs27 in-house datasets, the SSL F1 scores were comparable to Cellpose cyto2, but with significantly faster processing times, saving 25 minutes across over 500 images on the same CPU. For the 40X Hs27 images, SSL achieved a higher F1 score of 0.831 and a faster processing time compared to Cellpose cyto2, which also failed to segment approximately 12% of the images. A similar trend was observed for the 60X Hs27 images stained for F-actin and vinculin in which the cyto2 and cyto3 F1 scores were 0.608 and 0.454, respectively, in contrast to a significantly higher F1 score of 0.758 for SSL. In addition, the Cellpose models required more processing time while successfully segmenting only about half those images.

When evaluating a Cell Paint dataset of Hoechst-stained nuclei that was also used to train the Cellpose 'nuclei' model, Cellpose outperformed SSL, achieving a F1 score of 0.953 with 100% segmented in 58 minutes, versus a 0.873 SSL F1 score in 23.5 minutes. In general, the table shows that SSL demonstrates faster processing times and higher fidelity segmentation, particularly for the in-house Hs27 10X, 40X and 60X datasets, compared to Cellpose. Furthermore, as described above, SSL does not require that manual tuning of a diameter parameter for optimization.

Figure 8 presents representative image segmentations from Table 1 by SSL (Figures 8a, c, e) and Cellpose (Figures 8b, d and f). In most scenarios it is clear by eye that Cellpose has a tendency towards false negatives which lower its respective F1 scores in Table 1. The exception is with the nuclei segmentation,

where SSL tended towards over-segmentation (Figure 8e) compared to the 'nuclei' model of Cellpose (Figure 8f). Additionally, SSL counted surrounding flow signals near neighboring nuclei, reducing segmentation precision relative to Cellpose. However, as noted above, Cellpose was specifically trained on these images while still requiring the manual tuning of a diameter ($d = 30$) whereas SSL is self-trained and completely automated.

	Epifluorescent Hs27 10X		Phase Contrast Hs27 10X		Epifluorescence Hs27 40X		Epifluorescence Hs27 60X			Epifluorescence Nucleus	
Data source	in-house		in-house		in-house		in-house			Cell Painting	
Stains	F-actin, Vinculin		N/A		Vinculin		F-actin, Vinculin			Hoechst 33342	
No. of images	20		565		17		67			189	
Image Size	2304 × 2304		1344 × 1024		1324 × 1024		2262 × 2262			696 × 520	
Segmentation technique	SSL	Cellpose	SSL	Cellpose	SSL	Cellpose	SSL	Cellpose		SSL	Cellpose
Model *	-	cyto2	-	cyto2	-	cyto2	-	Cyto2	cyto3	-	nuclei
Diameter	-	100	-	100	-	150	-	400	400	-	30
F1	0.888	0.882	0.771	0.777	0.831	0.818	0.758	0.608	0.454	0.873	0.953
Times (mins)	10.48	16.67	123	148.05	3.02	5.15	19.04	33.78	40.53	23.45	58.00
% of exported segmentation images	100%	100%	100%	100%	100%	88.23%	100%	62.69%	53.73%	100%	100%

Table 1: Segmentation performance by SSL and an optimized Cellpose model for a range of imaging datasets. The table details information on data source, optical modalities, image size and required Cellpose inputs. Output metrics include F1 scores, processing times, and percentage of images segmented from the data set. d: diameter (in pixels), *Only required by Cellpose.

Figure 8: Representative image segmentation results from Table 1 by SSL and an optimized Cellpose model, (a) and (b) 10X Hs27 stained with Vinculin, (c) and (d) 60X Hs27 stained with F-actin and Vinculin, (e) and (f) nucleus stained with Hoechst. Scale bar: a-b: 50 μ m, c-d: 25 μ m, e-f: 20 μ m.

Supplemental Note 3: Summary of segmentation performance by SSL and different Cellpose models on several imaging datasets.

	Model	% of exported segmentation images	F1 score	Time (minutes)
Epifluorescence Hs27 60X Number of images = 67	SSL	100%	0.758	19.03
	cp_cyto (d=50)	82.09%	0.078	12.07
	cp_cyto (d=100)	49.25%	0.119	8.33
	cp_cyto (d=200)	38.81%	0.192	9.47
	cp_cyto (d=300)	29.85%	0.253	8.11
	cp_cyto (d=400)	35.82%	0.339	13.73
	cp_cyto2 (d=50)	100%	0.059	12.54
	cp_cyto2 (d=100)	85.07%	0.195	10.58
	cp_cyto2 (d=200)	82.09%	0.455	15.64
	cp_cyto2 (d=300)	68.66%	0.674	26.83
	cp_cyto2 (d=400)	62.69%	0.608	33.80
	cp_cyto3 (d=50)	86.57%	0.072	14.78
	cp_cyto3 (d=100)	70.15%	0.191	17.21
	cp_cyto3 (d=200)	74.63%	0.337	36.2
	cp_cyto3 (d=300)	67.16%	0.400	43.95
cp_cyto3 (d=400)	53.73%	0.454	38.63	
Epifluorescence Hs27 10X Number of images = 20	SSL	100%	0.8877	10.49
	cp_cyto2 (d = 50)	100%	0.795	11.05
	cp_cyto2 (d = 100)	100%	0.882	16.67
	cp_cyto2 (d = 150)	100%	0.792	16.56
Phase Contrast Hs27 10X Number of images = 565	SSL	100%	0.771	129.33
	cp_cyto2 (d = 30)	100%	0.622	142.00
	cp_cyto2 (d = 50)	100%	0.777	148.05
	cp_cyto2 (d = 100)	100%	0.614	118.7
Epifluorescence Nucleus Number of images = 189	SSL	100%	0.873	23.47
	cp_nuclei (d = 30)	100%	0.953	58.01
	SSL + cp_nuclei (d=30)	100%	0.944	76.03
	SSL + declumping	100%	0.873	24.22
Epifluorescence Hs27 40X Number of images = 17	SSL	100%	0.831	3.02
	cp_cyto2 (d=150)	82.35%	0.818	5.154
DIC S. cerevisiae 63X Number of images = 60	SSL	100%	N/A	36.35
	yeast_PhC_cp3 (d= 50)	100%	N/A	65.67
	SSL + declumping	100%	N/A	39.14

- For the supervised learning, the authors use Naive Bayesian Classifier. Did they try other classifiers?

In constructing our algorithm, we initially explored classifiers such as random forests, SVM, and K Nearest Neighbor. However, the Naïve Bayes classifier was chosen as a flexible and effective option, as they are known to have good bias-variance tradeoff because of their simplistic assumption of feature independence, and was found to perform robustly in the context of cell segmentation outlined here.

More details in:

COMMUNICATIONS BIOLOGY | <https://doi.org/10.1038/s42003-022-04117-x>

- *fully connected FPN should briefly described in the introduction.*

Corrected. Thank you.

- *The term "unstainable" should be replaced with "unsustainable" in the sentence "In addition, the increasing requirement of extracting fine cellular details from this imagery means that curated training data sets will need to grow exponentially at a rate that, in our view, is simply unstainable."*

Thank you for catching this.